# Screening of *Lentinula edodes* Strains for High Polysaccharide Production and In Vitro Antioxidant Activities

**DOI:** 10.3390/jof11050347

**Published:** 2025-04-30

**Authors:** Jie Zhang, Kanwal Rida, Jiahao Wen, Xiumei Yu, Yunfu Gu, Maoqiang He, Qiang Chen, Quanju Xiang

**Affiliations:** College of Resource, Sichuan Agricultural University, Chengdu 611130, China; 18708401916m@sina.cn (J.Z.); guyf@sicau.edu.cn (Y.G.); hemq@sicau.edu.cn (M.H.); cqiang@sicau.edu.cn (Q.C.)

**Keywords:** polysaccharide productions, antioxidant activity, polysaccharide components, molecular weight, enzyme activities

## Abstract

Lentinan is one of the main metabolites of *Lentinula edodes* and exhibits numerous biological properties, such as antitumor and antioxidant activity. Despite recent advancements, its commercialization remains constrained by a lengthy cultivation cycle, low yield, and high cost. Therefore, screening strains with high polysaccharide production or enhanced bioactivity at the mycelial fermentation stage is of significant importance. In this study, the mycelial polysaccharide content and in vitro antioxidant activity of 18 *L. edodes* strains were evaluated under shaking and static culture conditions. The total polysaccharide content and IC_50_ values under both culture conditions served as indicators for screening high-yielding and high in vitro antioxidant activity strains. Strain XG21 demonstrated superior polysaccharide production, with a total polysaccharide content of 78.80 mg in 50 mL of culture medium, which was 1.82 times higher than that of the main cultivated strain Xin808 (43.30 mg). Additionally, strain XG19 was identified for its high in vitro antioxidant activity, with total IC_50_ values of 3.11 and 3.38 mg mL^−1^ under shaking and static culture conditions, respectively. Further analyses on polysaccharide components, molecular weight, and enzyme activities were conducted on strains XG19, XG21, and Xin808. The results reveal that the polysaccharide from strain XG19 exhibited high uronic acid content and a significant weight-average molecular weight. Specifically, the intracellular polysaccharide uronic acid content (2.96%) was 2.22 and 1.14 times higher than that of Xin808 and XG21, respectively, while its weight-average molecular weight (Mw, 702.924 kDa) was 2.60 and 1.28 times greater than that of Xin808 and XG21. While the uronic acid content in its extracellular polysaccharides (EPSs) (8.26%) was similar to Xin808 and XG21, the Mw (83.894 kDa) was 1.56 times greater than that of XG21. Correlation analysis revealed that the content of extracellular polysaccharides and total polysaccharides was positively correlated with phosphoglucose isomerase (PGI) activity but negatively correlated with phosphoglucomutase (PGM) activity. These findings provide valuable strain information for the screening of mycelial polysaccharides with high yields and bioactivities.

## 1. Introduction

*Lentinula edodes*, commonly known as shiitake mushroom, is classified within the Basidiomycetes class, Agaricales order, and Tricholomataceae family under the genus *Lentinula*. This species is renowned for its high nutritional value, as it contains a wide range of essential amino acids, abundant mineral elements, and various trace elements [1,2]. Additionally, *L. edodes* is rich in various medicinal compounds, including lentinan, which exhibits a range of therapeutic properties. Lentinan has been shown to possess antitumor, anti-viral, and antioxidant effects, as well as the ability to lower blood pressure and blood lipid levels. Due to these beneficial properties, it has been widely utilized in clinical applications [3,4]. Lentinan, the main secondary metabolite of *L. edodes*, is a polysaccharide characterized by a β-(1→3)-D-glucan main chain. Traditionally, it is extracted from the fruiting bodies of the mushroom. However, the cultivation of fruiting bodies is a time-intensive and often inconsistent process, resulting in low lentinan yields and inefficient extraction of active compounds. These limitations hinder its broader development and application. Consequently, extracting and purifying active polysaccharides, such as lentinan, from mycelium through fermentation represents a highly promising and significant approach to overcome these challenges.

The selection of a strain capable of efficiently producing both biomass and bioactive compounds is crucial for optimizing the fermentation process. Microbial secondary metabolism is influenced by numerous factors, including strain characteristics [5,6] and fermentation conditions [7,8]. For instance, studies have shown significant variations in polysaccharide and ganoderic acid content among different *Ganoderma lucidum* strains, with strain-specific differences observed in the production of both intracellular and extracellular polysaccharides [5,6]. Fermentation techniques are primarily divided into two categories: static culture and shaking culture. These methods differ in cultivation approaches, oxygen supply, growth effects, and application scenarios. In liquid static culture, *G. lucidum* undergoes differentiation and morphological changes, leading to the formation of aerial mycelia and asexual spores, which are associated with high concentrations of ganoderic acids [9,10]. A two-stage fermentation approach, combining shaking culture followed by static culture, has been shown to enhance both the total triterpene content and yield [11]. However, whether differences exist in polysaccharide content and bioactivity among different *L. edodes* strains or under shaking and static fermentation conditions remains unclear. Understanding these variations is essential for optimizing the production of bioactive compounds in *L. edodes*.

Mushroom polysaccharides exhibit a wide range of biological activities, including notable antioxidant effects. These activities are influenced by a combination of factors, such as extraction and purification methods, polysaccharide structure, culture conditions, and the regulation of exogenous substances. For example, polysaccharides extracted from *L. edodes* at four different developmental stages demonstrated varying physicochemical properties and bioactivities. Among these stages, the immature phase was identified as the optimal harvest time for obtaining crude polysaccharides with higher biological activity [10]. Heteropolysaccharides derived from *Poria cocos* mycelia exhibit varying antitumor activities depending on the strain and culture medium used [12]. Similarly, polysaccharides extracted from *L. edodes* demonstrate significant antioxidant activity in a concentration-dependent manner [13,14]. For instance, at a concentration of 3.0 mg mL^−1^, the scavenging rate for DPPH radicals reached 98.47%, comparable to that of vitamin C [15]. The bioactivities of polysaccharides are influenced by structural characteristics such as monosaccharide composition, molecular weight distribution, degree of branching, and degree of sulfation [16,17]. Among these factors, monosaccharide composition and molecular weight are the most straightforward and commonly measured indicators of polysaccharide properties.

Therefore, in this study, the mycelial biomass, polysaccharide content, and in vitro antioxidant activity of 19 *L. edodes* strains were evaluated under both static and shaking submerged culture conditions, and strains with the highest polysaccharide content and antioxidant activity were screened. The screened strains were further analyzed for their polysaccharide components, molecular weight, and enzyme activity, aiming to establish a foundation for further understanding the relationships between polysaccharide synthesis, structural characteristics, and bioactivity.

## 2. Materials and Methods

### 2.1. Experimental Materials

Xin808, a commercially cultivated *L. edodes* strain, was obtained from the Chengdu Academy of Agricultural and Forestry Sciences. The remaining strains were generated by our group through monokaryotic hybridization in earlier stages, having undergone cultivation trials, and exhibited relatively high stability. For strain preservation and activation, Potato Sucrose Agar (PDA) medium was utilized, prepared with 200 g of potato, 20 g of glucose, 20 g of agar, and distilled water to make up 1 L. Three activated plugs (diameter 5 mm) were inoculated into 50 mL of fermentation medium, which consisted of 35 g of sucrose, 5 g of peptone, 2.5 g of yeast powder, 1 g of KH_2_PO_4_·H_2_O, 0.5 g of MgSO_4_·7H_2_O, and 0.5 g of vitamin B_1_ and was adjusted to pH 7.0, and diluted to 1 L with distilled water. The cultures were incubated under two conditions: shaking (initial static period of 1 day followed by shaking at 150 r min^−1^ for 39 days at 25 °C) and static (maintained at 25 °C for 40 days). Each treatment was performed in triplicate to ensure biological reproducibility.

### 2.2. Determination of Mycelial Biomass

The mycelium was separated from the liquid culture medium via filtration. The collected mycelium was rinsed three times with distilled water to remove any residual medium, followed by drying in an oven at 60 °C until a constant weight was achieved. The dry weight (g) of the mycelium from 50 mL of culture medium was then measured to determine the mycelial biomass.

### 2.3. Polysaccharide Extractions and Determinations

Polysaccharides were extracted using the hot water alcohol precipitation method as described previously [8]. Intracellular polysaccharides (IPSs) were extracted from the mycelium, while extracellular polysaccharides (EPSs) were obtained from the filtered fermentation broth. The polysaccharide content was quantified using the phenol–sulfuric acid method, with D-glucose serving as the standard [18]. The total IPS, total EPS, and total polysaccharide (TPS) content in 50 mL of fermentation broth was calculated to compare differences between different strains. The calculation methods were as follows: the total IPS content was calculated using the formula GI = Y × g, where GI is the total IPS content (mg), Y is the IPS concentration (mg g^−1^), and g is the mycelium biomass (mg); the total EPS content was calculated using the formula GE= w × v, where w is the EPS concentration (mg mL^−1^) and v is the volume of the fermentation broth (50 mL); and the TPS content was the sum of the total intracellular and total extracellular polysaccharides.

### 2.4. In Vitro Antioxidant Activities of Polysaccharides

The in vitro antioxidant activities of intracellular polysaccharides (IPSs) and extracellular polysaccharides (EPSs) were assessed based on their ability to scavenge the superoxide anion (O_2_^−^), hydroxyl (·OH), and 1,1-diphenyl-2-picrylhydrazyl (DPPH) radicals, following the methodology described by Xiang et al. [8]. Solutions of IPSs, EPSs and Vitamin C (used as a positive control) were prepared at varying concentrations (0.2, 0.4, 0.6, 0.8, and 1.0 mg mL^−1^) for the antioxidant activity assays. Deionized water, replacing the polysaccharide solution, served as the negative control. The IC_50_ value, defined as the concentration required to reduce 50% of the initial free radical concentration, was determined as per Liu et al. [19].

### 2.5. Enzyme Activities and Expression Profiles of Key Genes in Polysaccharide Synthesis

Fresh mycelium (0.1 g) was washed three times with 1 mL of 20 mM phosphate buffer (pH 6.5), ground into a fine powder using liquid nitrogen, and then homogenized in 1 mL of the same phosphate buffer. The homogenate was centrifuged at 10,000 rpm for 15 min at 4 °C, and the resulting supernatant was collected as the crude enzyme solution. The activities of key enzymes involved in polysaccharide synthesis, including phosphoglucose isomerase (PGI), phosphoglucomutase (PGM), and UDPG-pyrophosphorylase (UGP), were determined according to previously established methods [7]. Enzyme activity was quantified based on the molar extinction coefficient of NAD(P)H (ε340 = 6.22 × 10^3^ mol L^−1^ cm^−1^), with one unit of enzyme activity defined as the amount required to oxidize 1 nmol of NAD(P)H per minute.

Total RNA was extracted using the TRIzol method and subsequently reverse transcribed into cDNA with the AMV First-Strand cDNA Synthesis Kit (Sangon Biotech, Shanghai, China). The transcriptional expression levels of *PGI*, *PGM*, and *UGP* were quantified following the previously described protocol [7].

### 2.6. Determination of Polysaccharide Monosaccharide Components and Molecular Weight

The monosaccharide composition of the crude polysaccharides was analyzed following the method described by Salvador et al. and Zhu et al. [20,21]. Briefly, approximately 5 mg of the polysaccharide sample was hydrolyzed using 2 M trifluoroacetic acid (TFA) at 121 °C for 2 h in a sealed tube. The hydrolyzed sample was dried under a nitrogen stream, washed with methanol, and then dried again. This methanol washing step was repeated 2–3 times to ensure the complete removal of residual TFA. The resulting residue was re-dissolved in deionized water and filtered through a 0.22 μm microporous membrane for further analysis.

High-performance anion-exchange chromatography (HPAEC) was performed on a CarboPac PA-20 anion-exchange column (3 × 150 mm; Dionex, Sunnyvale, MA, USA) coupled with a pulsed amperometric detector (PAD; Dionex ICS 5000 system). The analysis conditions were as follows: flow rate, 0.5 mL min^−1^; injection volume, 5 μL; solvent system A (ddH_2_O), solvent system B (0.1 M NaOH), and solvent system C (0.1 M NaOH, 0.2 M NaAc). The gradient program was set as follows: 95:5:0 (A:B:C) at 0 min, 85:5:10 at 26 min, 85:5:10 at 42 min, 60:0:40 at 42.1 min, 60:40:0 at 52 min, 95:5:0 at 52.1 min, and 95:5:0 at 60 min. Data acquisition was performed using the ICS5000 system (Thermo Scientific, Waltham, MA, USA), and the results were processed with Chromeleon 7.2 CDS software (Thermo Scientific). The quantified data were exported into Excel (Version 2024) for further analysis and interpretation.

The molecular weight of the polysaccharides was measured according to the method reported previously [22,23]. The polysaccharide samples were dissolved in a 0.1M NaNO_3_ aqueous solution containing 0.02% NaN_3_ at a concentration of 1 mg mL^−1^ and filtered through a filter of 0.45 μm pore size. The homogeneity and molecular weight of the polysaccharide fractions were analyzed using size-exclusion chromatography coupled with multi-angle laser light scattering and refractive index detection (SEC-MALLS-RI).

The weight-average molecular weight (Mw), number-average molecular weight (Mn), and polydispersity index (Mw/Mn) were measured using a DAWN HELEOS-II laser photometer (Wyatt Technology Co., Santa Barbara, CA, USA) equipped with three tandem columns (300 × 8 mm, Shodex OH-pak SB-805, 804, and 803; Showa Denko K.K., Tokyo, Japan). The columns were maintained at 45 °C using a column heater. The elution flow rate was set at 0.4 mL min^−1^. A differential refractive index detector (Optilab T-rEX, Wyatt Technology Co., Santa Barbara, CA, USA) was connected in parallel to determine the concentration of the fractions and the specific refractive index increment (dn/dc). The dn/dc value for the polysaccharide fractions in the 0.1 M NaNO_3_ aqueous solution containing 0.02% NaN_3_ was determined to be 0.141 mL g^−1^. Data acquisition and processing were performed using ASTRA 6.1 software (Wyatt Technology). The quantified results were exported into Excel for further analysis and interpretation.

### 2.7. Data Analysis

The analysis was conducted using SPSS 24.0 software (SPSS Inc., Chicago, IL, USA). The Parson correlation coefficient (r^2^, ranging from +1 to −1) was calculated to analyze the relationship between the key enzyme activities involved in polysaccharide synthesis and the polysaccharide content from the data of the three varieties. Correlations were interpreted as follows: no correlation (r = 0), positive correlation (r > 0), or negative correlation (r < 0). All data are presented as means ± standard deviation (SD) derived from triplicate experiments. Statistical comparisons of means were performed using the least significant difference (LSD) test, with significance set at *p* ≤ 0.05.

## 3. Results

### 3.1. Polysaccharide Content of Mycelium from Different Strains Under Shaking and Static Culture Conditions

To identify the strain with the highest polysaccharide production, eighteen strains of L. edodes were inoculated into 50 mL of liquid medium and cultivated under both shaking and static fermentation conditions. The mycelial biomass, intracellular polysaccharide (IPS), extracellular polysaccharide (EPS), and total polysaccharide content were systematically measured and compared across all strains.

Among the eighteen strains cultivated, six strains exhibited mycelial biomass over 0.2 g in 50 mL of culture medium after 39 days of shaking culture. Strain XG24 demonstrated the highest mycelial biomass (0.55 g), which was 1.29 times greater than that of the reference strain *L. edodes* Xin808 (Appendix A). Additionally, seven strains showed intracellular polysaccharide (IPS) concentrations above 15 mg g^−1^, with strains XG20 and XG21 displaying the highest IPS levels at 21.75 mg g^−1^ and 17.97 mg g^−1^, respectively. Furthermore, three strains—XG21, XG24, and XG30—produced extracellular polysaccharide (EPS) concentrations exceeding 0.50 mg mL^−1^. Notably, strain XG21 exhibited the strongest EPS production capacity (0.70 mg mL^−1^), representing an 89.82% increase compared to Xin808 (Appendix A). When evaluating total polysaccharide content as an indicator, strain XG21 demonstrated superior polysaccharide production from shaking culture, with a total content of 44.06 mg, which was 88.31% higher than that of Xin808 (Table 1).

Similar to *G. lucidum*, which exhibits higher biomass under static culture conditions [9], *L. edodes* strains also demonstrate greater biomass accumulation under static culture conditions, and there were only four strains exhibiting a biomass below 0.20 g. UStrains XG14 and XG24 achieved the highest biomass, measuring 0.56 g and 0.57 g, respectively, which were 1.65 and 1.68 times greater than that of the reference strain Xin808 (Appendix A). Among the strains, 14 displayed intracellular polysaccharide (IPS) concentrations exceeding 15 mg g^−1^. Strains XG20 and XG21 demonstrated strong IPS and EPS production capabilities, with IPS and EPS contents reaching 74.94 mg g^−1^ and 0.51 mg mL^−1^, respectively. These values were 5.77 and 0.91 times higher than those of Xin808. When evaluating total polysaccharide content from static culture as an indicator, strain XG20 exhibited superior polysaccharide production under static culture, with a total content of 46.06 mg, representing a 1.31-fold increase over Xin808. Across the two culture conditions, strain XG21 emerged as the top performer in mycelial polysaccharide production, with a total polysaccharide content of 78.80 mg.

### 3.2. In Vitro Antioxidant Activity of Mycelium Polysaccharides from Different Strains Under Shaking and Static Culture Conditions

The antioxidant activities of mycelium intracellular polysaccharides (IPSs) and extracellular polysaccharides (EPSs) from 18 *L. edodes* strains were assessed using three distinct in vitro assays. As summarized in Appendix A, IPS and EPS derived from different strains under both static and shaking culture conditions demonstrated varying levels of antioxidant activity.

The mycelium IPSs and EPSs from strain XG5, cultured under shaking conditions, exhibited strong DPPH radical scavenging activity, with IC_50_ values of 0.01 and 0.20 mg mL^−1^, respectively (Appendix A). Similarly, under static culture conditions, the IPSs and EPSs from strain XG4 demonstrated notable DPPH radical clearance, with IC_50_ values of 0.01 and 0.20 mg mL^−1^, respectively (Appendix A). In terms of hydroxyl radical (OH) scavenging ability, the EPSs from strain XG5 under shaking culture and the IPSs from strain XG14 under static culture displayed lower IC_50_ values, measuring 0.07 and 0.01 mg mL^−1^, respectively. Additionally, IPSs and EPSs from strain XG19, cultured under shaking conditions, showed excellent superoxide anion (O_2_−) radical scavenging activity, with IC_50_ values of 0.01 and 0.19 mg mL^−1^, respectively. When considering the total IC_50_ values as the screening criterion, the polysaccharides from strain XG19 exhibited strong in vitro antioxidant activity, with total IC_50_ values of 3.11 and 3.38 mg mL^−1^ under shaking and static culture conditions, respectively (Table 2). Along with the main cultivated strain Xin808, strain XG21 (high polysaccharide yield) and strain XG19 (high in vitro antioxidant activity) were further analyzed for their polysaccharide synthase activity and structural characteristics.

### 3.3. Transcriptional Expression Levels and Enzyme Activities of Key Enzymes in Polysaccharide Biosynthesis

Polysaccharide biosynthesis is regulated by a complex network of enzymes, among which phosphoglucomutase (PGM), phosphoglucose isomerase (PGI), and UDPG-pyrophosphorylase (UGP) play pivotal roles [24]. The activities of these key enzymes in three *L. edodes* strains (high polysaccharide yield strain XG21, high in vitro antioxidant activity strain XG19, and Xin 808) are illustrated in Figure 1A. UGP exhibited the highest activity, while PGM showed the lowest. Strain Xin808 demonstrated the highest UGP activity (966.96 U mg^−1^), which was 1.9 and 5.1 times greater than that of strains XG19 and XG21, respectively. In contrast, strain XG21 displayed the highest PGI activity (275.50 U mg^−1^), surpassing that of Xin808 and XG19 by 3.1 and 3.0 times, respectively.

The transcriptional expression levels of three key enzymes in the three strains are presented in Figure 1B. The results reveal that the gene expression levels of *PGI* and *PGM* in strains XG19 and XG21 were lower than those in Xin808. Specifically, the expression level of *PGI* in Xin808 was 1.9 and 21.3 times higher than that in XG19 and XG21, respectively, while the expression level of *PGM* in Xin808 was 1.4 and 2.3 times higher than that in XG19 and XG21. In contrast, the highest expression of *UGP* was observed in strain XG19, being 6.0 and 23.2 times greater than that in Xin808 and XG21, respectively.

The Pearson correlation coefficient analysis between the key enzyme activities involved in polysaccharide synthesis and the polysaccharide content from the data of the three varieties indicated no significant correlation between intracellular polysaccharide (IPS) content and the activities of the three key enzymes (Table 3). However, extracellular polysaccharide (EPS) content showed a strong positive correlation with PGI activity (r = 0.981, *p* < 0.01) and a significant negative correlation with PGM activity (r = −0.851, *p* < 0.01). Additionally, the total polysaccharide content was positively correlated with PGI activity (r = 0.987, *p* < 0.01) and negatively correlated with PGM activity (r = −0.782, *p* < 0.05).

### 3.4. Monosaccharide Composition of Mycelial Polysaccharides

The structural features of polysaccharides play a critical role in determining and influencing their bioactivities. Monosaccharides, as the fundamental units and building blocks of polysaccharides, not only affect properties such as electrification, functional group composition, and bioactivity but also serve as one of the most straightforward and measurable indicators of polysaccharide structure [16]. To further investigate these relationships, we analyzed the monosaccharide composition of the three *L. edodes* strains (high polysaccharide yield strain XG21, high in vitro antioxidant activity strain XG19, and Xin 808).

Previous reports showed that the monosaccharide composition of polysaccharides varied within the same strain from different producing areas [24], in different parts [25], or among different strains of the same species [26]. This study further revealed differences in polysaccharide composition among the three strains examined, as well as distinct variations between intracellular and extracellular polysaccharides within the same strain. Eight monosaccharides—fucose, galactose, glucose, xylose, mannose, ribose, galacturonic acid, and glucuronic acid—were identified in the IPSs of the three *L. edodes* strains (Table 4). In contrast, ribose was absent from all EPS samples, and xylose was not detected in the EPS of strain XG21, indicating that only six monosaccharides were present in the EPS of XG21. Mannose was the most abundant monosaccharide, with EPSs containing a higher proportion of mannose (ranging from 51.41% to 51.91%) compared to IPSs (ranging from 30.65% to 35.11%). These findings align with previous studies showing that mannose, glucose, and galactose are common monosaccharides in mushrooms [27].

Notably, the IPSs from strain XG19 contained a higher proportion of galacturonic acid (1.22%), which was 2.18 and 1.39 times greater than that of Xin808 and XG21, respectively. This observation is consistent with research indicating that an increase in uronic acid content enhances the antioxidant activity of polysaccharides, as demonstrated in *Sagittaria sagittifolia* L. [28], and other studies showing a positive correlation between uronic acid content and antioxidant activity [29].

These data clearly demonstrate that *L. edodes* polysaccharides are heterogeneous, with mannose and galactose dominating the IPS composition, while mannose and glucose are predominant in EPSs. Arabinose (Ara) and galacturonic acid (GalA) appear to play critical roles in determining the IC_50_ values for DPPH-scavenging activity [30]. Interestingly, the IPSs from the three strains with lower galacturonic acid content exhibited better DPPH scavenging activity, suggesting a complex relationship between monosaccharide composition and antioxidant properties.

### 3.5. Analysis of Molecular Weight of Mycelial Polysaccharides

Molecular weight is a critical structural feature that influences the bioactivity of polysaccharides. Studies have shown that low-molecular weight (low-Mw) polysaccharides often exhibit superior immunomodulatory effects compared to their high-molecular weight (high-Mw) counterparts [31].

The molecular weight distributions of intracellular polysaccharides (IPSs) and extracellular polysaccharides (EPSs) from the three strains are summarized in Table 5. The weight-average molecular weight (Mw) and number-average molecular weight (Mn) of the IPSs were consistently higher than those of the EPSs. Among the strains, XG19 exhibited the highest molecular weights for both IPSs and EPSs. Specifically, the Mws of IPSs and EPSs from strain XG19 were 702.924 kDa and 83.894 kDa, respectively, which were 2.60 and 1.05 times greater than those of Xin808 and 1.28 and 1.56 times higher than those of XG21.

The weight-average molecular weight (Mw) of polysaccharides from strain Xin808 was higher than that of the other two strains. The IPSs and EPSs of strain Xin808 measured 69.401 kDa and 83.894 kDa, respectively, which were 2.33 and 1.87 times greater than those of XG19 and 2.37 and 3.168 times higher than those of XG21. The ratio of Mw to Mn (Mw/Mn), known as the polydispersity index (PDI), reflects the uniformity of polysaccharide molecular weight distribution. An analysis of the PDI values revealed that IPSs and EPSs from Xin808 exhibited smaller PDIs (3.887 and 9.01, respectively), indicating a more uniform molecular weight distribution compared to the other strains.

Polysaccharides with molecular weights ranging from 4 to 100 kDa are generally associated with high DPPH radical scavenging activity [30]. It is widely accepted that polysaccharides with lower molecular weights tend to exhibit stronger antioxidant activity [32,33]. In this study, the molecular weights of extracellular polysaccharides (EPSs) were consistently lower than those of intracellular polysaccharides (IPSs) across the three strains. This difference in molecular weight may partially explain the higher DPPH scavenging activity observed for EPSs compared to IPSs.

## 4. Conclusions

In this study, *L. edodes* strains XG21 and XG19 were identified as promising candidates due to their high polysaccharide productivity and strong in vitro antioxidant activity, respectively. Further analysis of polysaccharide components, molecular weight, and enzyme activities was conducted on XG19, XG21, and the widely cultivated strain Xin808. Strain XG19, which exhibited superior in vitro antioxidant activity, was characterized by higher uronic acid content and weight-average molecular weight (Mw). Specifically, the uronic acid content in its intracellular polysaccharides (IPSs) (2.96%) was 2.22 and 1.14 times higher than that of Xin808 and XG21, respectively, while its Mw (702.924 kDa) was 2.60 and 1.28 times greater than that of Xin808 and XG21. While the uronic acid content in its extracellular polysaccharides (EPS) (8.26%) was similar to Xin808 and XG21, the Mw (83.894 kDa) was1.56 times greater than that of XG21.

Correlation analysis revealed that the content of EPS and total polysaccharides (TPS) was positively correlated with phosphoglucose isomerase (PGI) activity and negatively correlated with phosphoglucomutase (PGM) activity. These findings suggest that strains XG19 and XG21 are potential candidates for further process optimization and scale-up studies aimed at enhancing their polysaccharide production and bioactivities. However, additional research is needed to test strain stability and optimize the fermentation conditions, extraction methods, and purification processes for these strains to maximize polysaccharide yield and bioactivity. Such efforts will help broaden the applications of *L. edodes* polysaccharides in various fields.

## Figures and Tables

**Figure 1 jof-11-00347-f001:**
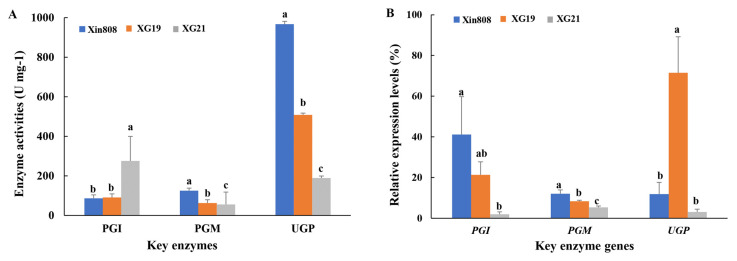
Enzyme activities (**A**) and relative expression levels (**B**) of key enzymes involved in polysaccharide biosynthesis in three *L. edodes* strains. Different lowercase letters above columns denote statistically significant differences among the strains at *p* < 0.05 (ANOVA-LSD, *n* = 3).

**Table 1 jof-11-00347-t001:** Total mycelial polysaccharide content of 18 *L. edodes* strains under shaking and static culture.

Strain No.		TPS (mg) ^a^	
	Shaking ^b^	Static ^c^	Shaking and Static ^d^
Xin808	23.40 ± 2.80 ef	19.90 ± 1.13 i	43.30 ± 2.24 h
XG2	14.77 ± 0.81 g	19.36 ± 0.29 i	34.14 ± 0.66 i
XG3	29.86 ± 2.23 cd	33.36 ± 0.91 c	63.23 ± 2.35 bcd
XG4	19.97 ± 2.36 f	27.95 ± 0.24 ef	47.92 ± 1.98 fg
XG5	12.28 ± 0.99 g	19.03 ± 1.15 i	31.31 ± 0.98 i
XG6	17.40 ± 0.56 fg	24.47 ± 0.5 g	41.86 ± 0.85 h
XG8	8.36 ± 0.43 h	9.23 ± 0.2 k	17.59 ± 0.34 j
XG9	27.29 ± 2.66 de	32.64 ± 0.41 c	59.93 ± 2.20 de
XG12	20.27 ± 0.29 f	11.01 ± 0.18 j	31.28 ± 0.36 i
XG13	25.65 ± 1.43 e	35.28 ± 0.75 b	60.93 ± 1.45 cde
XG14	14.90 ± 0.17 g	28.92 ± 0.93 e	43.82 ± 0.89 gh
XG19	25.70 ± 1.43 e	24.14 ± 0.24 g	49.85 ± 1.30 f
XG20	29.15 ± 1.84 d	46.06 ± 0.41 a	75.21 ± 1.83 a
XG21	44.06 ± 2.69 a	34.74 ± 0.47 b	78.80 ± 2.58 a
XG22	18.18 ± 2.24 f	21.22 ± 0.58 h	39.40 ± 1.36 h
XG24	37.54 ± 4.15 b	27.53 ± 0.21 f	65.07 ± 3.54 bc
XG26	27.44 ± 2.04 de	30.07 ± 0.4 d	57.51 ± 1.65 e
XG30	32.39 ± 1.58 c	33.34 ± 0.27 c	65.72 ± 1.51 b

^a^: TPS content is the sum of intracellular and extracellular polysaccharides extracted from 50 mL of fermentation broth and calculated by the formular as shown in Section 2.3; ^b^: the TPS content from the shaking culture; ^c^: the TPS content from the static culture; ^d^: the TPS from the shaking and static culture. All values are presented as the means of three replicates ± standard deviation (SD). Different lowercase letters adjacent to the values indicate statistically significant differences between individual strains at *p* < 0.05, as determined by ANOVA followed by the least significant difference (LSD) test (*n* = 3). This notation applies to all subsequent data presented in a similar manner.

**Table 2 jof-11-00347-t002:** The total IC_50_ values of antioxidant activities of polysaccharide from 18 strains.

Strain No.	Total IC_50_ (mg mL^−1^) ^a^		
	Shaking ^b^	Static ^c^	Shaking and Static ^d^
Xin808	27.79 ± 2.64 c	15.06 ± 2.60 h	42.85
XG2	18.26 ± 1.38 f	8.63 ± 1.08 j	26.89
XG3	53.99 ± 3.83 a	23.22 ± 2.71 d	77.21
XG4	5.83 ± 0.31 j	3.48 ± 0.31 no	9.31
XG5	3.21 ± 0.19 m	19.13 ± 0.83 f	22.34
XG6	33.23 ± 3.55 b	4.16 ± 0.23 l	37.39
XG8	4.77 ± 0.34 k	37.54 ± 2.34 b	42.31
XG9	6.18 ± 0.97 j	4.28 ± 1.11 l	10.46
XG12	3.21 ± 0.25 m	15.76 ± 0.48 g	18.97
XG13	23.45 ± 2.61 e	65.29 ± 5.82 a	88.74
XG14	4.05 ± 0.30 l	3.82 ± 0.79 m	7.87
XG19	3.11 ± 0.21 m	3.38 ± 0.14 o	6.49
XG20	3.11 ± 0.39 m	4.11 ± 0.46 l	7.22
XG21	7.10 ± 1.40 i	8.99 ± 0.12 i	16.09
XG22	26.90 ± 2.42 d	37.36 ± 3.27 c	64.26
XG24	7.99 ± 0.26 h	3.57 ± 0.11 n	11.56
XG26	4.01 ± 0.36 l	22.24 ± 3.37 e	26.25
XG30	15.86 ± 1.74 g	5.86 ± 1.17 k	21.72

^a^: the total IC_50_ value is the sum of intracellular and extracellular polysaccharide IC_50_ value; ^b^: the total IC_50_ value of IPSs and EPSs from the shaking culture; ^c^, the total IC_50_ value of IPS and EPS from the static culture; ^d^: the total IC_50_ from the shaking and static culture. Different lowercase letters above columns denote statistically significant differences among the strains at *p* < 0.05 (ANOVA-LSD, *n* = 3).

**Table 3 jof-11-00347-t003:** Correlation analysis between polysaccharide content and enzyme activity.

Enzyme Activity	IPS	EPS	TPS
PGI	0.426	0.981 **	0.987 **
PGM	0.125	−0.851 **	−0.782 *
UGP	0.455	−0.603	−0.533

The Pearson correlation coefficients were evaluated between the contents and enzyme activity from the data of the three varieties. “**” and “*” indicate significant (*p* < 0.05) and highly significant (*p* < 0.01) correlations, respectively.

**Table 4 jof-11-00347-t004:** Monosaccharide composition and percentage of mycelium IPS and EPS of three *L. edodes* strains.

Monosaccharide Composition	Monosaccharide Percentage (%)					
	Xin 808		XG19		XG21	
	IPS	EPS	IPS	EPS	IPS	EPS
Fucose	6.00	3.50	4.04	1.93	2.66	1.07
Galactose	30.32	11.16	30.90	6.74	31.43	6.25
Glucose	20.62	23.06	24.19	29.77	28.09	33.08
Xylose	5.55	2.79	2.22	1.38	3.24	-
Mannose	35.11	51.41	34.59	51.91	30.65	51.65
Ribose	1.06	-	1.11	-	1.33	-
Galacturonic acid	0.56	2.11	1.22	2.43	0.88	2.22
Glucuronic acid	0.77	5.97	1.74	5.83	1.72	5.74

“-”, not detected.

**Table 5 jof-11-00347-t005:** Molecular weight of mycelium IPSs and EPSs of three *L. edodes* strains.

Molecular Weight	Xin 808		XG19		XG21	
	IPS	EPS	IPS	EPS	IPS	EPS
Mw (kDa)	269.783	79.875	702.924	83.894	547.913	53.824
Mn (kDa)	69.401	8.865	29.752	4.750	29.229	3.168
Mz(kDa)	1780.179	2375.146	6448.187	5629.469	3551.611	1500.835
PDI (Mw/Mn)	3.887	9.010	23.626	17.663	18.746	16.992

## Data Availability

The original contributions presented in this study are included in the article/Appendix A. Further inquiries can be directed to the corresponding author.

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
