# Peer review of "Screening of Lentinula edodes Strains for High Polysaccharide Production and In Vitro Antioxidant Activities"

_jof, 2025, doi:10.3390/jof11050347_

Round 1
Reviewer 1 Report
Authors should add discussion. Page 8: How can you explain that the monosaccharide composition of polysaccharides varies among strains of the same species?
Latin names, as well as “in vitro”, should be italicized in the text. Check the correctness of reference 18. The difference in uronic acid content by 1.02 and 1.05 times is within the error limits of the method.
Author Response
Dear Reviewer,
Thank you for assessing our manuscript, and for the helpful comments. We have acceptedyou’re your valuable suggestions. Colour coding: in “Response to Reviewers”, Referee comments are in black, our responses are in blue; in “Revised manuscript”, we have highlighted the relevant changes in red. A detailed response can be found below.
Quanju Xiang, on behalf of all authors
-----------------------------------------------
Point-by-point responses to the comments from Reviewer 1:
[comments]In the Introduction, the authors write that polysaccharides are mainly extracted from fruit bodies. It is necessary to compare the yield when extracting from fruit bodies and vegetative mycelium. Are the same polysaccharides extracted from fruit bodies and vegetative mycelium? Moreover, the authors write «…polysaccharides extracted from L. edodes at four different developmental stages demonstrated varying physicochemical properties and bioactivities”. What is known? What is known about the static and shaking culture of Lentinus? With Ganoderma, things may be different.
Response:Thanks for your valuable comments. The polysaccharides extracted from fruit bodies and vegetative mycelium are different, since the biosynthesis, structure, and activity of polysaccharides vary significantly across different growth stages and environmental conditions, making them highly susceptible to environmental influences. Polysaccharides derived from the fruiting bodies and mycelium stages of Lentinula edodes also exhibit distinct characteristics. While commercially available polysaccharides are primarily extracted from fruiting bodies, their acquisition is time-consuming and inconsistent. Consequently, exploring mycelium-derived polysaccharides with comparable bioactivities to those from fruiting bodies, along with optimizing fermentation conditions for such strains, has emerged as a novel research direction in polysaccharide applications.
We are sorry for the confusion. The citation “…. polysaccharides extracted from L. edodes at four different developmental stages demonstrated varying physicochemical properties and bioactivities” aims to demonstrate that polysaccharides are influenced by multiple factors, including growth stages. static and shaking culture are two different culture methods, which may also influence polysaccharides.
[comments] The authors describe their data, but there is no comparison with the already known data and no good discussion. It is unclear whether the polysaccharides isolated by the authors are lentinan? It is necessary to clearly show what the authors have done new in comparison with the known data on the polysaccharides of Lentinus and other basidiomycetes.
Response:Thanks for your comments. Lentinan is the biologically active polysaccharide derived from Lentinula edodes, and the screening criteria used in this study was the crude polysaccharide content, thus the general term "polysaccharide" was consistently applied throughout the text. In the revised manuscript, additional discussions have been added.
[comments] Figure 1 needs to be redone. It's very small now, hard to see. Table 1 is titled "Molecular Weight..." but contains data on the monosaccharide composition.
Response: Thanks, the figure and table have been modified in the revised manuscript.
[comments] Authors should add discussion. Page 8: How can you explain that the monosaccharide composition of polysaccharides varies among strains of the same species?
Response:Thanks. The structure of polysaccharides is influenced by factors such as sources, extraction methods, and chemical modification techniques. Significant differences may exist even within the same strain at different growth stages, in different parts, or among different strains of the same species. Such structural differences can consequently affect their bioactivity. The relevant literatures have been incorporated into the revised manuscript.
[comments] Latin names, as well as “in vitro”, should be italicized in the text. Check the correctness of reference 18. The difference in uronic acid content by 1.02 and 1.05 times is within the error limits of the method.
Response:Thanks for your comments. All the Latin names and “in vitro” has been corrected in the revised manuscript. As the reviewer pointed out that “1.02 and 1.05 times is within the error limits of the method”, this misleading statement has been eliminated from the revised manuscript.
Reviewer 2 Report
Please see attached file
Please see attached file

Author Response
Dear Reviewer,
Thank you for assessing our manuscript, and for the helpful comments. We have acceptedyou’re your helpful suggestions. Colour coding: in “Response to Reviewers”, Referee comments are in black, our responses are in blue; in “Revised manuscript”, we have highlighted the relevant changes in red. A detailed response can be found below.
Quanju Xiang, on behalf of all authors
-----------------------------------------------
Point-by-point responses to the comments from Reviewer 1:
Abstract
Comments: You wrote: "Strain XG21 demonstrated superior polysaccharide production, with a total polysaccharide content of 78.80 mg, which was 1.82 times higher than that of the main cultivated strain Xin808 (43.3 mg)". *How is concentration measured? mg/L, mg/ml, of the fermentation solution? Or mg/gr of the mycelium (dry or wet)?
Response: Thanks for your comment. The total polysaccharides represent the sum of intracellular and extracellular polysaccharides. And the detailed calculation for intracellular and extracellular polysaccharides is provided in Section “2.3 Polysaccharide extractions and determinations”.
Introduction
Comments: You wrote: "Ganoderma lucidum". *It is usually written Ganoderma lucidum and lucidum.
You wrote: "Poria cocos.". *It is usually written: Poria cocos
Response: We sincerely appreciate the reviewers’ comments. The Latin names of species should be italicized, and we have carefully checked and revised them throughout the revised manuscript.
Comments: You wrote: ". Among these, strains XG21 and XG19 exhibited higher polysaccharide yields and superior antioxidant activity" It is not acceptable to refer to the results of the present research in the introductory section of the article. The research methods and results should be detailed first and the results later.
Response: Thanks for your valuable comments. This part has been revised as “…, and strains with the highest polysaccharide content and antioxidant activity are screened. Alongside the widely cultivated strain Xin808, further analyses were con-ducted to characterize the polysaccharide components, molecular weight distribution, and enzyme activities.” in the revised manuscript.
Materials and methods
Comments: You wrote: "the remaining strains were generated through monosporal hybridization to ensure heterozygosity."*Were these strains created by you for this research? How long had these strains existed before the experiments began? How can we be sure that these are stable strains that can be used for cultivation?
Response: Thanks. Our research group has been engaged in the breeding of L. edodes, primarily focusing on the agronomic traits of fruiting bodies. These hybrids were previously obtained by our group through monokaryotic hybridization and have undergone fruiting experiments, demonstrating stable mushroom production. Later, we aimed to develop L. edodes varieties for deep processing, e.g. polysaccharide production. Given the long cultivation cycle of fruiting bodies and the poor stability of agricultural cultivation methods, we thus decided to initiate screening at the mycelial stage.
Results
Comments: You wrote: “Among the nineteen strains cultivated, six exhibited a mycelial biomass exceeding 0.20 g after 39 days…..” *Are these results in g/L of the fermentation medium or something else? You continue to give such weight results without indicating if they represent content in the growth medium or in the (dry/wet) mycelium? You wrote: Table 1. Mycelial polysaccharide content of L. edodes strains under shaking and static culture.*Above and under this table there is no explanation what are the numbers in the table? Do they represent polysaccharide yield in mg/L of the medium or mg/g of the mycelium?
Response: Thanks for pointing out this. The biomass is the total dry mycelium biomass in 50 mL of fermentation, in order to make it clear, this part has been modified as “six strains exhibited mycelial biomass over 0.2 g in 50 mL culture medium after 39 days of shaking culture”, and the detailed methods are described in Section 2.2.
Both the intracellular polysaccharide (IPS) and extracellular polysaccharide (EPS) are determined, and the total polysaccharides represent the sum of intracellular and extracellular polysaccharides. The data in Table 1 represent the total polysaccharide content, the detailed content for IPS and EPS are provided in Table S1. To enhance clarity, we have revised both the table names and notes." in the revised manuscript.
Comments: Starting from section 3.3 in the results chapter, you present results for only three strains: the commercial strain and two of the new strains. I did not find any clear indication that highlights why only these two new strains were chosen to be presented the results?
Response: Thanks. We are sorry to make this confusion. Since only strains with high polysaccharide content (XG21) and strong in vitro antioxidant activity (xg19) were chosen for further analysis, together with strain 808 (the main cultivated variety) serving as the control. A statement “Along with the main cultivated strain Xin808, strain XG21 (high polysaccharide yield) and strain XG19 (high in vitro antioxidant activity) will be analyzed for their polysaccharide synthase activity and structural characteristics.” has been added in the revised manuscript.
Comments: *Fig.1. It is customary to write the units of measurement of the results presented in each figure below the figure. I did not find them in this figure, although they are presented on the side of fig. 1A.
Response: Thanks for pointing out this. Figure 1 has been redone.
Comments: *Table 3: The correlation analysis between polysaccharide content and enzyme activity was done for one fungal variety, or average of the three varieties?
Response: Thanks. The relationship between the key enzyme activities involved in polysaccharide synthesis and the polysaccharide content was done from the data of the three varieties, and the modification has been added in “2.7 Data Analysis”.
Comments: Table 5. The title is not clear enough. In addition, the table itself includes the contents of Table 4. So, in fact Table 5. is not displayed at all. This is a major problem because it is impossible to understand and relate to the molecular weight data you are discussing.
Response: We sincerely appreciate the reviewers for identifying the issues. We sincerely apologize for the error that occurred while using the journal template, and the correct table has been replaced in the revised manuscript.
Comments: Conclusions
You wrote: "These findings suggest that strains XG19 and XG21 are potential candidates for further process optimization and scale-up…" I agree, but first you need to make sure they are stable strains that are suitable for commercial use.
Response: Thank you for the reviewer's comments. As the reviewers pointed out, the selected strains must exhibit stability for further commercial utilization. Based on the current research findings, we will conduct further analyses, including stability testing, determination of polysaccharide content in fruiting bodies. And a statement “However, additional research is needed to test strain stability, optimize fermentation conditions” has been modified in the revised manuscript.

Round 2
Reviewer 1 Report
No comments
No comments
Author Response
We sincerely appreciate your valuable time and expertise in reviewing the manuscript.
Reviewer 2 Report
The article is generally fine. See my attached comments which also include some needs for necessary clarifications.
Please see the attached file. It includes my comments on the first version of the article, to which I added what was corrected correctly (which is the majority) and what still requires correction or clarification.

Author Response
Comments1: You wrote: ". Among these, strains XG21 and XG19 exhibited higher polysaccharide yields and superior antioxidant activity" It is not acceptable to refer to the results of the present research in the introductory section of the article. The research methods and results should be detailed first and the results later.
You left this part in the introduction, contrary to my opinion. The editor will decide about it.
Response: Thanks for your valuable comments. We sincerely apologize for having misinterpreted the requirement regarding the inclusion of preliminary screening results in the Introduction section. Consequently, we have removed those results and in the revised manuscript, this section has been revised to concisely outline the research objectives and significance of this study.
Results
Comments2: You wrote: Table 1. Mycelial polysaccharide content of edodes strains under shaking and static culture. *Above and under this table there is no explanation what are the numbers in the table?
Do they represent polysaccharide yield in mg/L of the medium or mg/g of the mycelium?
You added details below the table, but there is still no answer to my question (see above).
Response: hanks for your valuable comments, and we are sorry for not explaining it clearly. The table name and table notes have been modified as “Table 1. Total mycelial polysaccharide content of 18 L. edodes strains under shaking and static culture”, “TPS content is the sum of intracellular and extracellular polysaccharides extracted from 50 mL of fermentation broth and calculated by the formulars as shown in Section 2.3” to make it clearer.
Due to variations in biomass, intracellular polysaccharide concentration, and extracellular polysaccharide concentration among different strains, we evaluated the polysaccharide production capability of the strains using total polysaccharides (TPS, the sum of total intracellular and total extracellular polysaccharides). The intracellular polysaccharide (IPS) was extracted from the mycelium and the total intracellular polysaccharide is the calculated by the formula GI=Y*g, where GI is the total IPS content (mg), Y is the IPS concentration (mg g-1), g is the mycelium biomass (mg), while extracellular polysaccharide (EPS) was extracted from the fermentation broth and the total extracellular polysaccharide is the calculated by the formula GE= w*v, where w is the EPS concentration (mg mL-1), and v is the volume of the fermentation broth (50 mL). The detailed statements are shown in the Materials and Methods section 2.3 (please see P3, line 122-129).
Comments3: *Table 3: The correlation analysis between polysaccharide content and enzyme activity was done for one fungal variety, or average of the three varieties?
I still don’t find reply to this question.
Response: Thanks. The relationship between the key enzyme activities involved in polysaccharide synthesis and the polysaccharide content was done from the data of the three varieties, and the statement has been modified as “The Parson correlation coefficient (r2, ranging from +1 to −1) was calculated to analyze the relationship between the key enzyme activities involved in polysaccharide synthesis and the polysaccharide content from the data of the three varieties.” in the revised manuscript.(Please see P5, line 194-196)
